# Tackling Multi-Physics Nano-Scale Phenomena in Capillary Force Lithography with Small Data by Hybrid Intelligence

**DOI:** 10.3390/mi14111984

**Published:** 2023-10-26

**Authors:** Ashish Chapagain, In Ho Cho

**Affiliations:** Department of Civil, Construction, and Environmental Engineering, Iowa State University, Ames, IA 50011, USA; cashish@iastate.edu

**Keywords:** nano capillary lithography, nano-grating, transparent machine learning, Bayesian evolutionary algorithm, hybrid intelligence

## Abstract

The scientific community has been looking for novel approaches to develop nanostructures inspired by nature. However, due to the complicated processes involved, controlling the height of these nanostructures is challenging. Nanoscale capillary force lithography (CFL) is one way to use a photopolymer and alter its properties by exposing it to ultraviolet radiation. Nonetheless, the working mechanism of CFL is not fully understood due to a lack of enough information and first principles. One of these obscure behaviors is the sudden jump phenomenon—the sudden change in the height of the photopolymer depending on the UV exposure time and height of nano-grating (based on experimental data). This paper uses known physical principles alongside artificial intelligence to uncover the unknown physical principles responsible for the sudden jump phenomenon. The results showed promising results in identifying air diffusivity, dynamic viscosity, surface tension, and electric potential as the previously unknown physical principles that collectively explain the sudden jump phenomenon.

## 1. Introduction

Nanoscale structures are found commonly in nature and serve various essential functions. For instance, a chameleon’s skin contains nanoscale crystals, which allows them to modify their appearance by adjusting their spacing. Learning from nature, modern progress in nanotechnology is geared towards harnessing the potential of nanotextures to perform a wide array of functions by manipulating the height of nano pixels [1,2,3,4,5]. Nanoscale capillary force lithography (CFL) is a nanofabrication technique that cures a photopolymer using UV radiation [6] to create patterns and structures on the nanoscale. CFL is gaining attention due to its high precision, potential for mass production, and relatively low cost. However, the physical principles at play are still obscure. One observed phenomenon in experiments is the sudden jump, where the height of the photopolymer experiences a rapid change at a specific UV dose (Figure 1c) [6].

Several research studies have been carried out in this sphere to explain the capillary behavior in nanochannels [7,8,9,10]. One intriguing phenomenon investigated is the Klingenberg effect [11] and its incorporation in gas flow through porous media [12,13,14]. One of the most recent advances involved discovering the hidden rules that explain the sudden jump phenomenon, with gas diffusivity, dynamic viscosity, and surface tension identified as the previously obscure physical principles governing this phenomenon [15]. However, it could not find the relationship between the height of the PDMS mold and its effect on the sudden jump. This paper has attempted to overcome this drawback and explain the sudden rise phenomenon for all heights of PDMS mold by combining physical principles alongside artificial intelligence, known as “hybrid intelligence”. The hybrid intelligence approach is explained in Figure 1. 

Current machine-learning (ML) approaches [16,17,18] show promising outcomes [19,20,21]. However, they are “black-box” models where the inner working of the model is not easily understandable or interpretable. Due to the inability of ML to unveil “transparent rules”, this paper applies hybrid intelligence, used in [15], to explain the sudden jump phenomenon and its variation with PDMS height. Hybrid intelligence combines human and artificial intelligence to reveal the unknown physical rules (Figure 1d). On one hand, human intelligence draws upon fundamental physics, scientists’ observation, experience, and knowledge. Unknown terms that are beyond current human understanding are included in the formulation. On the other hand, artificial intelligence provides the search capabilities to identify the most promising expressions for these unknown rules [15].

Although this paper has its beginnings in [15], considerable advances have been made since. There are significant differences between this paper and [15]. Firstly, this paper focuses on nano-grating geometry, while [15] deals with circular nanopillars. This distinction necessitates substantial adjustments in the formulation based on human intelligence. Secondly, this paper includes four different nano-grating heights, and we collectively employ all four experiments for rule learning. In contrast, ref. [15] conducted rule learning using just one nano-pillar experiment. Thirdly, this paper has made further advances to [15] by incorporating electric potential difference as an additional physical process involved in the capillary rise in the nano-grating, thus requiring expansion of the previous three physics processes, i.e., dynamic viscosity, surface tension, and diffusivity [15].

This paper focuses on a few experiments (thus, small data) that commonly exhibit the complex sudden jump phenomena. The fundamental difficulty stems from the small length scale, time-varying salient properties (e.g., air pressure, air mass, diffusivity into porous medium, dynamic viscosity, surface tension, and electric potential-dependent forces), and the multi-physical phenomenon interactions during the nano capillary rising. Obtaining large data sets is often difficult and expensive. Sometimes, internal data are intrinsically inaccessible due to technological limitations such as the inability to monitor the liquid’s behavior inside the nano-ridge during the capillary rise and to precisely measure the electric charge inside the nano-ridge.

In our study, we observed four key phenomena that collectively contribute to the explanation of the sudden jump phenomenon:Air diffusivity decreases as the rate of trapped air-mass transfer increases;The dynamic viscosity of the liquid increases upon exposure to UV radiation;Our results demonstrate that UV exposure leads to a reduction in the surface tension of the liquid;The electric potential between the liquid and the PDMS mold decreases as the distance between the liquid’s top surface and the bottom surface of the PDMS mold increases.

These observed phenomena collectively contribute to our understanding of the sudden jump phenomenon, shedding light on the underlying physical principles at play in this intriguing phenomenon.

## 2. Materials and Methods

### 2.1. Data Preparation

We assessed the topography of the printed nano-ridge array using atomic force microscopy (AFM) in tapping mode, employing highly doped silicon tips (NCHR, NanoWorld, Neuchâtel, Switzerland). Comprehensive details regarding nanofabrication procedures and AFM data acquisition are outside the scope of this paper and will be the subject of a dedicated future publication. It should be noted that the total data points are artificially increased for smooth training using interpolation when the trend drastically changes, such as near the sudden jump region. Such regions show infinite slopes, which may cause mathematical issues and difficulty fitting with smooth LFs (Figure 1c). For instance, in the 1.5 μm case, two points (x, y) = (15,600, 1.35) and (15,600, 0.679) showing the infinite slope may be refined to five points (non-infinite slope) with additional points at (x, y) = (15,500, 1.35), (15,550, 1.18), (15,600, 1.0155), (15,650, 0.847), and (15,700, 0.679) using interpolation. Since a simple interpolation generates the increased data points, the overall shape of the experimental values does not change. All experimental data will be made available upon request to the author.

### 2.2. Two Pillars of the Hybrid Intelligence—Human and Artificial Intelligences

#### 2.2.1. Human Intelligence for Providing Basic Formulations with Basic Physics Quantities

The mathematical formulation used in this study is derived mainly from [15]. The formulation that relates the PDMS mold’s height with sudden jump—the potential (voltage) difference—is shown in the succeeding section. This section provides a comprehensive formulation and terminology governing the principles of force equilibrium and mass balance in the context of nano capillary rise.

Force equilibrium formulation

The ascent of a fluid within a vertical nano-ridge is determined using an interplay of multiple force components, which can be expressed as follows: $F_{s}-F_{v}-F_{P}-F_{g}+F_{e}=0$. Here, $F_{s}$ denotes the capillary force driven by the liquid’s surface tension, $F_{v}$ is the viscous force that counteracts the liquid’s rise, $F_{P}$ is the force due to air pressure confined within the nano-ridge [15], and $F_{g}$ is the gravitational force. A new force term is introduced into the formulation by [15], $F_{e}$, which characterizes the electric potential-dependent force. Figure 2 illustrates the liquid’s height x(t) at time t. The individual force components are defined as $F_{s}={2l\sigma }_{s}cos(\theta )$, $F_{v}=-12lx\left(t\right)\frac{\mu }{d}\frac{\delta x}{\delta t}$ [22], and $F_{p}=d\cdot l\left(P\left(t\right)-P\left(0\right)\right)$. For force due to electric potential, we considered the pressure caused by the electric force, denoted as $p_{e}$, which is described in [23]. The exact formulation is given by: $p_{e}=\frac{1}{2}\epsilon_{0}\left(\frac{1}{\epsilon_{1}}-\frac{1}{\epsilon_{2}}\right){\left[\frac{\epsilon_{1}\epsilon_{2}U}{\epsilon_{1}h+\epsilon_{2}\left(d-h\right)}\right]}^{2}$. Here, $\epsilon_{0}$ is the permittivity of a vacuum, $\epsilon_{1}$ is the dielectric constant of air, $\epsilon_{2}$ is the dielectric constant of the liquid, and U represents the applied potential difference. From this equation, it is evident that the pressure is directly proportional to the square of potential difference ($p_{e}\propto U^{2}$) and inversely proportional to the square of (d−h) ($p_{e}\propto \frac{1}{{\left(d-h\right)}^{2}}$). However, U, $\epsilon_{0},\epsilon_{1},$ and $\epsilon_{2}$ are unknown quantities. As a result, we introduce a term, U(h), which collectively accounts for the influence of these four variables. This leads to the establishment of the simplified expression for $p_{e}$ as $p_{e}$= $\frac{U\left(h\right)}{{\left(d-h\right)}^{2}}$. So, $F_{e}$ is the area on which the pressure exerts times the pressure force. Therefore, $F_{e}=\frac{d\cdot l\cdot U\left(h\right)}{{\left(h-x\left(t\right)\right)}^{2}}$, where all the unknown terms are contained in U(h), which will be learned by machine learning later. While we did not consider curvature for electric forces in this study, we agree that it is an important factor. We plan to include it in our future research to increase accuracy. Here, l is the length of the nano-ridge, $\sigma \left(U_{v}\right)$ [N/m] is the surface tension of the liquid (initial value = 0.04 N/m) which depends on the total UV dose ($U_{v}$) [$\frac{J}{m^{2}}$], θ [deg] is the contact angle between the liquid and the nano-ridge (75 degrees), $\mu \left(U_{v}\right)$ [$\frac{Ns}{m^{2}}$] is the $U_{v}$-dependent dynamic viscosity of the liquid (initial value = 0.4 $\frac{Ns}{m^{2}}$), $\frac{dx}{dt}$ is the average velocity of the rise of the liquid, d is the distance between two walls of the nano-ridge (1500 nm), P(t) [$\frac{N}{m^{2}}$] is the pressure at time t of the confined air, P(0), the initial air pressure, is assumed to be the atmospheric pressure (101,325 Pa), and g is the acceleration due to gravity. It is worth noting that the gravitational force is substantially smaller in magnitude compared to the other terms, rendering it negligible in the subsequent consideration. We arrive at the following expression upon incorporating all these force components into the equilibrium equation. It needs to be noted that while the formulation method is similar to [15], the following formulation differs in the geometry and inclusion of the electric force.
(1)$$2l\sigma_{s}cos\left(\theta \right)-12lx\left(t\right)\frac{\mu }{d}\frac{\delta x}{\delta t}-d\cdot l\cdot \left(P\left(t\right)-P\left(0\right)\right)+\frac{d\cdot l\cdot U\left(h\right)}{{\left(h-x\left(t\right)\right)}^{2}}=0.$$

Applying the ideal gas law to the air confined within the pressure term can be expressed in the following manner: $P\left(t\right)=\frac{m\left(t\right)}{Mdl\left(h-x\left(t\right)\right)}RT$ and $P\left(0\right)=\frac{m\left(0\right)}{Mdlh}RT$, where $m\left(t\right)$ [kg] is the mass of confined air at time t, R is the universal gas constant (8.3144 J/mol/K), T [K] is the absolute temperature (78 K), and M [kg/mol] is the molar mass of the confined air (0.02897 kg/mol). Substituting the pressure expressions into the equilibrium equation, we can derive the equation governing the velocity of the liquid’s ascent as follows:(2)$$\frac{\delta x}{\delta t}=\frac{\sigma \left(U_{v}\right)dcos\left(\theta \right)}{6\mu \left(U_{v}\right)x\left(t\right)}-\frac{RTd}{12\mu \left(U_{v}\right)lMx\left(t\right)}\left(\frac{m\left(t\right)}{h-x\left(t\right)}-\frac{m\left(0\right)}{h}\right)+\frac{d^{2}U\left(h\right)}{12\mu \left(U_{v}\right)x\left(t\right){\left(h-x\left(t\right)\right)}^{2}}$$
where m(t) is the mass of trapped gas at time t, h is the total height of the capillary (1090 nm, 1500 nm, 2000 nm, and 2500 nm), m(0) is the initial mass of the trapped gas, and U(h) is a link function to account for the voltage (potential) difference and dielectric constants, which will be learned by the proposed machine-learning method.

The time required for the liquid to rise to height (h) when the nano-ridge is open-ended (i.e., no cap) can derived by $t_{rise}$=$\int_{0}^{h}{\left(\frac{dx}{dt}\right)}^{-1}ds\;$= $\frac{3\mu h^{2}}{\sigma dcos\left(\theta \right)}$ [15]. 

2.Mass balance formulation

As air is trapped within the nano-ridge, it will gradually decrease due to diffusion into nanopores within the PDMS mold. To establish the rate of change of mass of the trapped gas, we can derive it from the mass balance equation as follows [15]:(3)$$\frac{\delta m}{\delta t}=\frac{-D_{d}K_{H}A\left(t\right)}{l_{cr}}\frac{RT}{d\cdot l}\left(\frac{m\left(t\right)}{h-x\left(t\right)}-\frac{m\left(0\right)}{h}\right)$$
where $D_{d}$ is the coefficient of diffusion, $K_{H}$ is Henry’s constant, $A\left(t\right)=d\cdot l+2l\left(h-x\left(t\right)\right)$ is the area through which diffusion takes place, and the distance at which the air pressure diffuses is taken as $l_{cr}$. Here $,l_{cr}$ is the nanopore pressure critical length [m] measured from the nano-ridge’s wall. At $l_{cr},$ the internal pore pressure becomes $p_{0}=P\left(0\right)$ (Figure 2). In this work $l_{cr}$ is assumed to be constant at 10h. $D_{K}\left(\frac{\;d\bar{m}}{d\bar{t}}\right)\equiv D_{d}K_{H}$ is a function of the air mass flux rate whose initial value is $1\times 10^{-12}$ mol·s/kg. $D_{d}$ is the diffusion coefficient of the air into PDMS pores. This study posits that a sudden surge in airflow into PDMS nanopores can impede air diffusion within these nanopores, much like a traffic jam can be caused by a sudden increase in traffic volume at a fixed intersection. Figure 1h illustrates this physical rationale schematically. Obtaining accurate time-varying models for $D_{d}$ and $K_{H}$ separately can be challenging. Therefore, this study employs a glass-box approach to identify the combined effect of $D_{d}$ and $K_{H}$, represented as $D_{K}$. The rise in height, mass, and time are normalized as
(4)$$\bar{x}\left(\bar{t}\right)=\frac{x\left(t\right)}{h};\;\bar{m}\left(\bar{t}\right)=\frac{m\;\left(t\right)}{m\left(0\right)};\;\bar{t}=\frac{t}{t_{rise}}.$$

Thus, normalized force equilibrium and mass balance equations are given as
(5)$$\frac{\delta \bar{x}}{\delta \bar{t}}=\frac{1}{2\bar{x}\left(\bar{t}\right)}-\frac{\alpha }{2}\left(\frac{\;\bar{m}\left(\bar{t}\right)-1+\bar{x}\left(\bar{t}\right)}{\;\bar{x}\left(\bar{t}\right)\left(1-\;\bar{x}\left(\bar{t}\right)\right)}\right)+\frac{\gamma }{\;\bar{x}\left(\bar{t}\right){\left(1-\bar{x}\left(\bar{t}\right)\right)}^{2}}$$
and
(6)$$\frac{\delta \bar{m}}{\delta \bar{t}}=-\beta \left(1+\frac{2h}{d}\left(1-\bar{x}\left(\bar{t}\right)\right)\right)\left(\frac{\;\bar{m}\left(\bar{t}\right)-1+\bar{x}\left(\bar{t}\right)}{\left(1-\bar{x}\left(\bar{t}\right)\right)}\right).$$

To simplify and condense the equations above, we introduce additional coefficients α, β, and γ. Here, $\alpha =\frac{P\left(0\right)d}{2\sigma cos\left(\theta \right)}$, $\beta =\frac{3\mu D_{d}K_{H}RTh}{d\sigma cos\left(\theta \right)l_{cr}}$, $\gamma =\frac{U\left(h\right)d}{4\sigma h^{2}cos\left(\theta \right)}$, and $D_{d}K_{H}$ is a function of $\frac{\delta \bar{m}}{\delta \bar{t}}$.

To this point, human intelligence mathematically formulates the liquid rise velocity and the mass rate in the presence of physics quantities. But they are not complete. Those physics-driven formulations inevitably contain several unknown terms, of which rules need to be tackled by artificial intelligence, as described in the following section. The physical phenomena under consideration and the corresponding link functions are summarized in Table 1.

#### 2.2.2. Artificial Intelligence for Exploring and Searching Hidden Rules

Flexible link functions

The search for transparent link functions (LFs) is conducted using the Bayesian evolutionary algorithm. For the effective utilization of LF (L), the input arguments have been normalized to the range [0, 1]. As in [15], the relationship between the UV dose and the terms σ and μ is related as $\sigma \left(U_{v}\right)=\sigma_{0}L_{\sigma }\left(\bar{U_{v}};\theta_{\sigma }\right)\;$ and $\mu \left(U_{v}\right)=\mu_{0}L_{\mu }\left(\bar{U_{v}};\theta_{\mu }\right)$, where $\sigma_{0}$ and $\mu_{0}$ are the surface tension and the dynamic viscosity of the liquid in the absence of UV exposure, respectively. The UV dose is normalized as $\bar{U_{v}}=U_{v}/36,000\;[J/m^{2}]\in R[0,1)$. θ is the free parameter vector, which is learned using the Bayesian algorithm. The initial values of $\sigma_{0}=0.04\;N/m$ and $\mu_{0}=0.4\;N$·$s/m^{2}$ is assumed for the study. The function $D_{K}\left(\frac{d\bar{m}}{d\bar{t}}\right)={D_{K0}L}_{D_{K}}\left(\underset{\forall t}{\max}\left[\frac{d\bar{m}}{d\bar{t}}\right];\theta_{D_{K}}\right)$ is used to describe the maximum flux rate over time, where $D_{K0}$ is the initial value of $D_{d}K_{H}$, and this study assumed $D_{d}K_{H}=1\times 10^{-12}$.

A relationship between the normalized height of grating ($\bar{h}$) and the physics term U will be related by function as $U\left(\bar{h}\right)=U_{0}L_{U}(\bar{h};\theta_{U})$, where $\bar{h}$ is normalized as $\frac{h}{10^{-6}}$. In this study, the initial value of $U_{0}$ is assumed $2\times 10^{-8}$.$\;L_{U}$ is a link function used to account for the potential (voltage) difference and dielectric constants.

A general two-parameter exponential LF [24,25,26,27,28], i.e., two-parameter exponential LF with the form $L\left(x\right)=\exp\left(c_{1}x^{c_{2}}\right)$, was chosen (Figure 3) for surface tension, dynamic viscosity, and diffusivity of air. To include the effect of electric force based on [23], a second-order polynomial LF of the form $L\left(x\right)=c_{1}x^{2}+c_{2}x+c_{3}$ was chosen among a few choices. The rationale for this selection of the LF is presented in the discussion section later. 

2.Bayesian evolutionary algorithm

This paper has employed a fusion of the genetic algorithm’s fitness-proportionate probability (FPP) rule and the Bayesian update scheme to ensure seamless rule learning. Following the FPP rule, the likelihood of an organism (denoted as “o”) from the present generation being chosen for the subsequent generation is directly related to its fitness score (F) where organism “o” is a distinct manifestation of free parameters $\left(\Theta =\left\{\theta_{1},\ldots ,\theta_{n_{rule}}\right\}\right)$ within all concealed rules. The fitness score (F) is given by $F\left(s\right)={\left(1+Err\left(s\right)\right)}^{-1}$ where $Err\left(s\right)=n^{-1}\sum_{i}^{n}\frac{{\left(x_{real}^{\left(i\right)}-x_{pred}^{\left(i\right)}\right)}^{2}}{{\left(x_{real}^{\left(i\right)}\right)}^{2}}$. A reduced error corresponds to increased fitness. The fitness score of the previous best generation and the collection of all the free parameter sets **Θ** from the same generation are represented as $F^{*}\left(o\right)$ and $O^{*}\left(\Theta \right),$ respectively. Since each **Θ** can be uniquely represented by an organism, the interchangeability of organism and **Θ** is established: p(o)∝F(o), equivalently $p\left(\Theta \right)\propto F\left(o\right)$. Subsequently, the Bayesian fitness score of a novel individual organism (denoted as $F_{B}\left(o\right)$) is formulated as $F_{B}\left(o\right)=\frac{1}{\kappa }\frac{F\left(o;\;O^{*}\left(\Theta \right)\right)F^{*}\left(o\right)}{\sum_{\forall o}F^{*}\left(o\right)}$ where $\kappa =\sum_{\forall o}\frac{F\left(o;\;O^{*}\left(\Theta \right)\right)F^{*}\left(o\right)}{\sum_{\forall o}F^{*}\left(o\right)}$ is a normalization term. The probability of parent selection for the upcoming generation is denoted as $p(parent_{i}\;|\;o)\propto F_{B}(o),(i=1,2)$. Consequently, all the identified LFs can seamlessly evolve with new experimental data. Within the Bayesian evolutionary framework, a cumulative count of 100,000 organisms and 10 generations are employed, encompassing four alleles for every gene. The mutation mechanism operates per-variable, with a mutation rate of 0.005. 

We can anticipate whether $L\left(x\right)$ will rise or fall using human intelligence. Figure 3 provides us with cues to deduce the signs of $c_{1}$ and $c_{2}$. Subsequently, we initiated our exploration within a broad interval. In our specific case, the initial ranges for $c_{1}$, $c_{2}$, and $c_{3}$ were set to a wide search range (ranging from −10 to +10). During the training process, we progressively narrowed down our search ranges using a technique known as the shrinking search range method, ultimately establishing our final search ranges. Therefore, hybrid intelligence found the best-performing search ranges $c_{1}\in \left[-8,-6\right]$ and $c_{2}\in \left[0,\;10\right]$ for LFs of diffusivity, $c_{1}\in \left[8,\;10\right]$ and $c_{2}\in \left[0,\;10\right]$ for LFs of dynamic viscosity, $c_{1}\in \left[-6,-4\right]$ and $c_{2}\in \left[0,\;10\right]$ for LFs of surface tension, and for LF of potential (voltage) difference, $c_{1}\in \left[0,\;0.15\right]$, $c_{2}\in \left[-0.4,\;0\right]$, and $c_{3}\in \left[0,\;1.3\right]$.

The model was trained using the Bayesian update, which updates free parameters (**Θ**) with each experimental case, subsequently following the order presented in Figure 4. In our model, we set $n_{test}$ = 4 and $n_{epoch}$ = 10. An epoch concluded after processing the 2.5 case, at which point the 1.09 case experimental data was reintroduced for further training. This process is iterated for up to 10 epochs. The model’s accuracy increases with more training epochs (Figure 5), but after the fifth epoch, accuracy appears to reach a plateau. Thus, this paper presents the best-so-far rules obtained after the fifth epoch of Bayesian evolutions.

### 2.3. Feasibility Test Result

The experiment consisted of four cases depending on the height of the grating. The heights were 1.09 μm, 1.5 μm, 2.0 μm, and 2.5 μm. The corresponding experiments are named the 1.09 case, 1.5 case, 2.0 case, and 2.5 case, respectively. Figure 6a–d compare experimental results of the nano capillary rising heights and reproductions with the best-so-far rules identified by the proposed hybrid intelligence approach. At the sudden change in the height near the total UV dose (15,000–17,000 $J/m^{2}$), we conducted AFM analyses on the samples right above and below the sudden jump for more quantitative comparisons. Applying the glass-box ML-identified rules, the final heights above and below the sudden jump were well reproduced (Figure 6a–d). Figure 6a–d reproduce the normalized height with increasing UV dose. The normalized height is the final height of liquid scaled down to a range of [0, 1] by $\frac{x\left(t\right)}{h}$. For instance, in the 1.09 case, normalized height = 1 means that the liquid has reached the final height of 1.09 μm.

Figure 7a–d show the mean and one-standard range of the reproduction of the liquid rise using the top 10 best-so-far rules for the four cases, respectively. This plot shows the uncertainty levels of the best-so-far rules’ prediction.

The transparent ML sought to learn four hidden rules (Figure 8)—i.e., two rules about the dynamic viscosity (Figure 8b) and surface tension (Figure 8c) of the photopolymer (NOA73) as nonlinear functions of the total UV doses; a rule about the air diffusivity (Figure 8a) as a nonlinear function of air mass flux rate into the nanopores in the mold made of PDMS; and a rule about the electric potential difference (Figure 8d).

## 3. Discussion

### 3.1. Parametric Study

We conducted parametric studies to understand the impact of the salient physics terms’ initial values—i.e., the initial values of the air diffusivity ${D_{d}K}_{H}$ (Figure 9), the dynamic viscosity μ (Figure 10), the surface tension σ (Figure 11), and the initial potential (voltage) difference U (Figure 12). As anticipated, the larger diffusivity results in the rapid rise of the liquid height, but the path is highly nonlinear (Figure 9). A similar trend is found for the higher dynamic viscosity (Figure 10). The initial electric potential appears to accelerate the liquid rise, but the trend is nonlinear (Figure 12). The normalized height ($\bar{x})$ is the representation of the liquid rise scaled between the range [0, 1] given by $\frac{x\left(t\right)}{h}$, where $\bar{x}$ = 1 means the liquid has attained its maximum height. Similarly, normalized mass ($\bar{m}$) is the mass of trapped air scaled between the range [0, 1] given by $\frac{m(t)}{m\left(0\right)}$. $\bar{m}$ = 1 signifies that no trapped air has escaped the capillary, and conversely, $\bar{m}$ = 0 signifies that all the trapped air has escaped through the PDMS pores. And normalized pressure ($\bar{p}$) = 1 signifies the actual pressure of the trapped air is equal to the atmospheric pressure.

As the diffusivity of the trapped air into the PDMS pores decreases, the time required for the liquid to hit the top appears to increase (Figure 9a). This is aligned with basic physics, i.e., the higher the diffusivity, the faster the trapped air escapes. Similarly, with higher diffusivity, the normalized mass ($\bar{m}$) decreases rapidly (Figure 9b) and trapped air pressure increases rapidly (Figure 9c).

Figure 10 shows the complex relationship between the rate of liquid rise and dynamic viscosity (μ). Equation (6) shows that $\frac{\delta \bar{m}}{\delta \bar{t}}$ ∝−β where $\beta =\frac{3\mu D_{d}K_{H}RTh}{d\sigma \cos\left(\theta \right)l_{cr}}$. This leads to $\frac{\delta \bar{m}}{\delta \bar{t}}$ ∝−μ. The second term of Equation (5) shows that $\frac{\delta \bar{x}}{\delta \bar{t}}$ ∝ −$\bar{m}(\bar{t})$. So, we can conclude $\frac{d\bar{x}}{d\bar{t}}\propto \mu ,$ which can be found in Figure 10a.

Figure 11 depicts a complex relationship between the rate of liquid rise and surface tension (σ). In the section discussing the formulation of mass balance, we introduced α, β,  and γ as follows: $\alpha =\frac{P\left(0\right)d}{2\sigma \cos\left(\theta \right)},\beta =\frac{3\mu D_{d}K_{H}RTh}{d\sigma \cos\left(\theta \right)l_{cr}},\;\gamma =\frac{U\left(h\right)d}{4\sigma h^{2}cos(\theta )\;}$. Consequently, it becomes evident that α is inversely proportional to σ ($\alpha \propto \frac{1}{\sigma }$), β is also inversely proportional to σ ($\beta \propto \frac{1}{\sigma })$, and γ follows the same trend, being inversely proportional to σ ($\gamma \propto \frac{1}{\sigma }$). Furthermore, Equation (5) indicates that as γ increases, $\frac{\delta \bar{x}}{\delta \bar{t}}$ also increases, while it decreases with increase in α. On the other hand, Equation (6) establishes that as β increases, $\frac{\delta \bar{m}}{\delta \bar{t}}$ also increases. The second term of Equation (5) also shows that $\frac{\delta \bar{x}}{\delta \bar{t}}$ ∝ −$\bar{m}(\bar{t})$. Consequently, surface tension exerts both positive and negative influence on the rate of liquid rise. Specifically, as surface tension increases, the time required for the liquid to attain its maximum height increases.

Figure 12 gives another complex relationship between the rate of liquid rise and potential (voltage) difference (U). A potential difference existing between the lower surface of the PDMS and the upper surface of the liquid generates an attractive force between them. Consequently, U actively contributes to the rate of liquid rise. Following Equation (5), it becomes evident that $\frac{\delta \bar{x}}{\delta \bar{t}}\propto \gamma$, where $\gamma =\frac{U\left(h\right)d}{4\sigma h^{2}cos(\theta )\;}.$ As the value of $U\left(h\right)$ increases, the rate of liquid rise experiences a corresponding increase, leading to a faster attainment of the liquid’s maximum height.

The link functions for diffusivity, surface tension, dynamic viscosity, and their impact on the rate of liquid rise and trapped-air mass reduction rate are derived from prior work [15]. These relationships, including the impact of potential (voltage) difference, are expressed through Equations (5) and (6), where *α*, *β*, and *γ* play key roles. The rate of liquid rise, $\left(\frac{\delta \bar{x}}{\delta \bar{t}}\right)$, is given by $\frac{\delta \bar{x}}{\delta \bar{t}}=\frac{1}{2\bar{x}\left(\bar{t}\right)}-\frac{\alpha }{2}\left(\frac{\;\bar{m}\left(\bar{t}\right)-1+\bar{x}\left(\bar{t}\right)}{\;\bar{x}\left(\bar{t}\right)\left(1-\bar{x}\left(\bar{t}\right)\right)}\right)+\frac{\gamma }{\;\bar{x}\left(\bar{t}\right){\left(1-\bar{x}\left(\bar{t}\right)\right)}^{2}}$, and the trapped-air mass reduction rate, $\frac{\delta \bar{m}}{\delta \bar{t}}=-\beta \left(1+\frac{2h}{d}\;\left(1-\bar{x}\left(\bar{t}\right)\right)\right)\left(\frac{\;\bar{m}\left(\bar{t}\right)-1+\bar{x}\left(\bar{t}\right)}{\left(1-\bar{x}\left(\bar{t}\right)\right)}\right)$ where, $\alpha =\frac{P\left(0\right)d}{2\sigma_{0}L_{\sigma }cos(\theta )}$, $\beta =\frac{3\mu_{0}{L_{\mu }\left(D_{d}K_{H}\right)}_{0}\;L_{DK}RTh}{d\sigma_{0}L_{\sigma }\left(x\right)cos(\theta ){\;l}_{cr}}$, and $\gamma =\frac{U_{0}L_{U}d}{4\sigma_{0}L_{\sigma }\left(x\right)h^{2}cos(\theta )}$. Notably, these link functions are interrelated, as evidenced by Equations (5) and (6). They collectively influence the height of liquid rise, demonstrating the intricate connection between diffusivity, surface tension, dynamic viscosity, and their effects on liquid behavior.

### 3.2. Comparisons among Different LFs for Electric Potential-Dependent LF

For diffusivity, surface tension, and dynamic viscosity, [15] explored various link functions, including constant, linear, simple nonlinear, complex nonlinear, and exponential forms. Upon examining the final height vs. UV dose graph, it was evident that the best fit was achieved with the expression $L\left(x\right)=\exp\left(c_{1}x^{c_{2}}\right)$ [15].

To explain the impact of the different types of LFs, two types of LFs were considered to account for the effects of potential (voltage) difference and dielectric constants. The first was an exponential function of the form $L_{U}\left(x\right)=\exp\left(c_{1}x^{c_{2}}\right)$ and the second was a second-order polynomial function of the form $L_{U}\left(x\right)=c_{1}x^{2}+c_{2}x+c_{3}$. 

Two different approaches were taken for the polynomial LF; in the first, the free parameters for all four link functions were found using the Bayesian evolution, and in the second, the free parameters for $L_{U}$ were kept constant, and parameters for the other three link functions were found. Using this method, we obtained four predictions for each experimental case after training the model for five epochs. The Mean Absolute Percentage Error (MAE) for each case was obtained (Figure 13), which showed that the polynomial LF where all the free parameters were found using the Bayesian evolution was the most accurate in predicting the sudden jump phenomenon.

The two best prediction results were from two cases of the polynomial LF with 26.39% and 13.10% MAE after training the model for five epochs. Therefore, the prediction result of the two was compared (Figure 14a,b). Figure 14b depicts the prediction results with the second-order polynomial LF with an increasing trend, which shows the sudden jump occurred at the same UV dose, which contradicts the experimental data. Conversely, Figure 14a (the decreasing trend of $L_{U}$) clearly captures the trend followed by the sudden jump. Based on the result of Figure 14, the best prediction was obtained using a second-order polynomial LF with decreasing trend of $L_{U}$.

### 3.3. Remarks on the Difference from Statistical Learning

The proposed hybrid intelligence approach is more general than the statistical approach. As shown in Figure 15, the hybrid intelligence systematically explores the diverse candidates of input-parameter pairs x→y where x∈ {UV dose, mass flux rate, maximum height of nano-gratings} and y∈ {diffusivity, viscosity, surface tension, electric potential} in pursuit of the most plausible rule shapes. Due to observational and technological limits, y are scarce, rendering direct statistical fitting infeasible. To some extent, the internal parameters (y) can be regarded as latent variables in the “encoder” ML methods [19,20,21] (Figure 15). Figure 15f–h show preliminary investigations about how bad-performing rule shapes are rejected during the learning.

## 4. Conclusions

This paper combines human insights with the searching power of machine learning to uncover hidden rules, collectively known as hybrid intelligence. This approach helped us understand the rules that explain the physical phenomenon of a sudden jump in nano-grating. Dynamic viscosity, diffusivity, and surface tension were previously recognized as the underlying physical phenomena explaining this occurrence. However, experimental data revealed the existence of a phenomenon that required an explanation for the correlation between the height of the PDMS mold and the final liquid height, which was unaddressed in prior research. Our hybrid intelligence approach allowed us to identify a potential (voltage) difference-dependent force that could fully elucidate the relationship between the mold’s height and the final height reached, thereby explaining the phenomenon, irrespective of the PDMS mold’s height.

## Figures and Tables

**Figure 1 micromachines-14-01984-f001:**
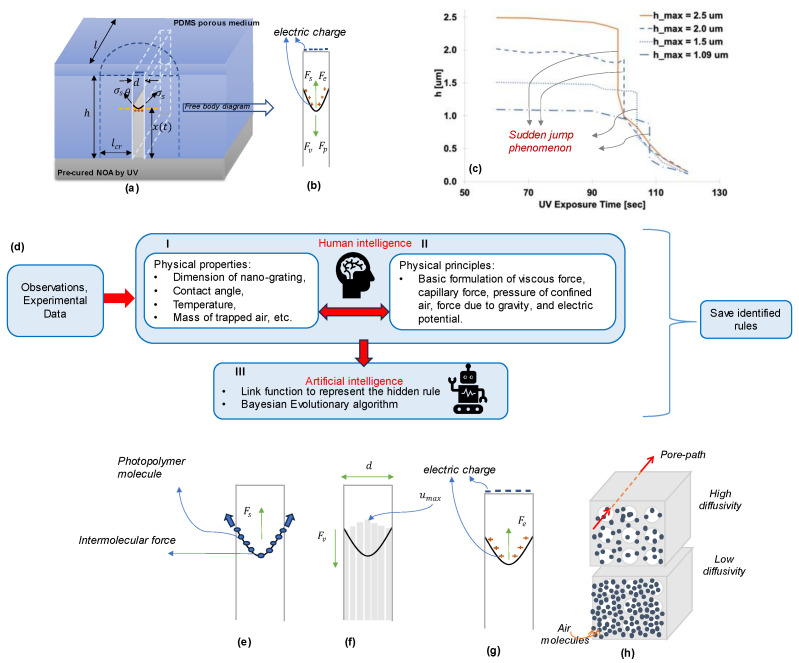
(**a**) Basic physics terms and geometrical properties of the capillary rising in a nano-grating and (**b**) their force equilibrium; (**c**) Height plots obtained from the rise of the same photopolymer into nano-grating with varying maximum heights. All cases commonly exhibit the sudden jump phenomena; (**d**) Architecture of the proposed hybrid intelligence approach. Step I includes the known parameters, which are included as input in our research. Step II consists of the known physical principles and their mathematical formulation. Steps I and II together make up human intelligence. Step III is the artificial intelligence, which includes a link function to identify the unknown physical phenomenon and Bayesian evolutionary algorithm; (**e**–**h**) Graphical representation of the recognized physical rules regarding surface tension (**e**), dynamic viscosity (**f**), electric (voltage) potential (**g**), and diffusivity of air molecules (**h**).

**Figure 2 micromachines-14-01984-f002:**
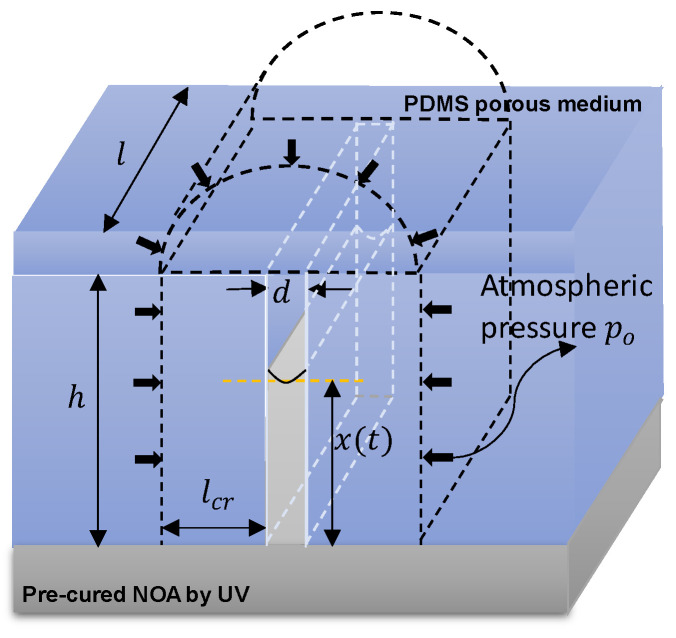
3D view of a single nano-ridge with pressure boundary conditions and geometry.

**Figure 3 micromachines-14-01984-f003:**
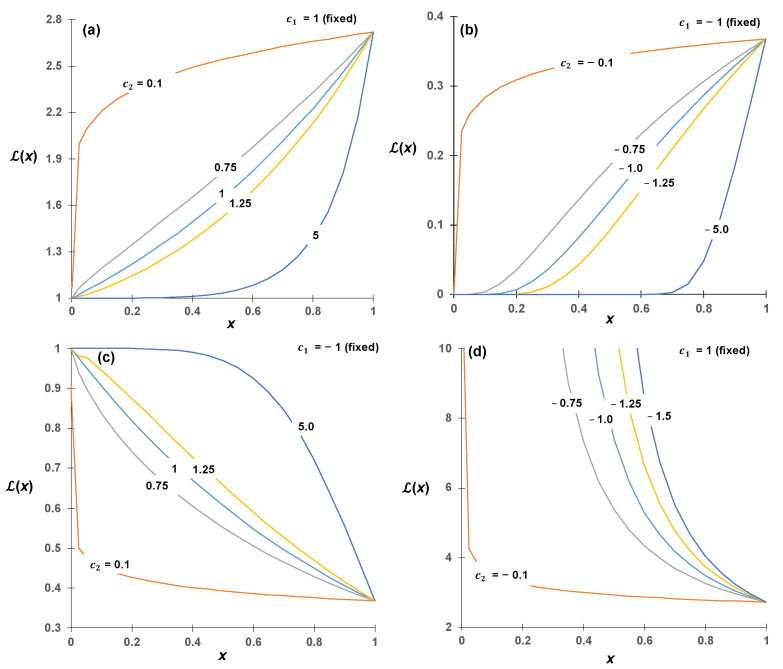
Example plots of the two-parameter exponential link function: general nonlinear increasing relations (**a**,**b**) and decreasing relations (**c**,**d**) where $L\left(x\right)=\exp\left(c_{1}x^{c_{2}}\right)$.

**Figure 4 micromachines-14-01984-f004:**
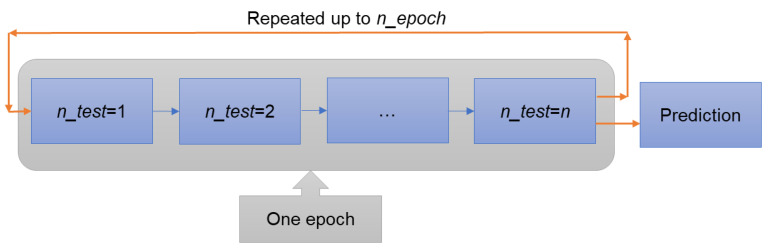
Bayesian evolutionary training sequence with multiple experiments.

**Figure 5 micromachines-14-01984-f005:**
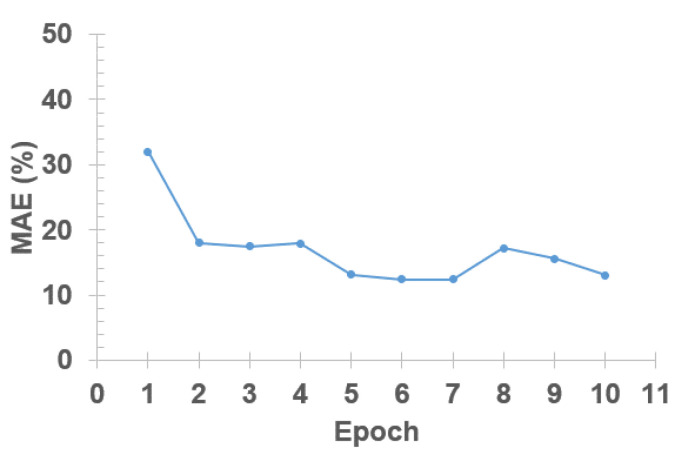
Accuracy variations of the Bayesian evolution with increasing training epochs.

**Figure 6 micromachines-14-01984-f006:**
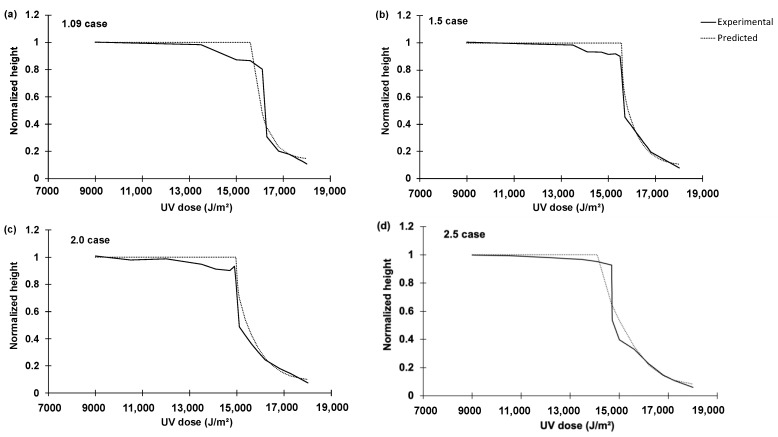
Comparison between experimental results and reproduction (prediction) of final height of nano capillary rising using UV-cured NOA73 with the best-so-far ML-identified rules for: (**a**) 1.09 case; (**b**) 1.5 case; (**c**) 2.0 case; and (**d**) 2.5 case. In all cases, the sudden jumps experimentally observed near $15,000\;\sim 17,000\;J/m^{2}$ are well captured by the transparent ML-identified rules.

**Figure 7 micromachines-14-01984-f007:**
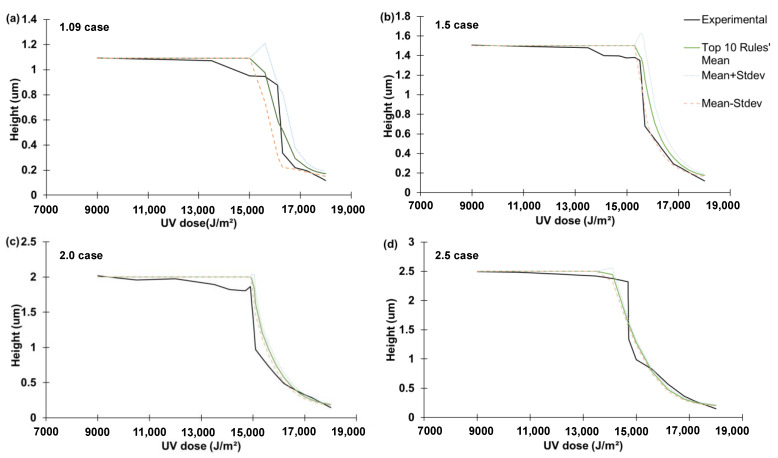
The mean and one standard deviation (Stdev) range of the reproductions using the top 10 ML-identified rules for the: (**a**) 1.09 case; (**b**) 1.5 case; (**c**) 2.0 case; and (**d**) 2.5 case.

**Figure 8 micromachines-14-01984-f008:**
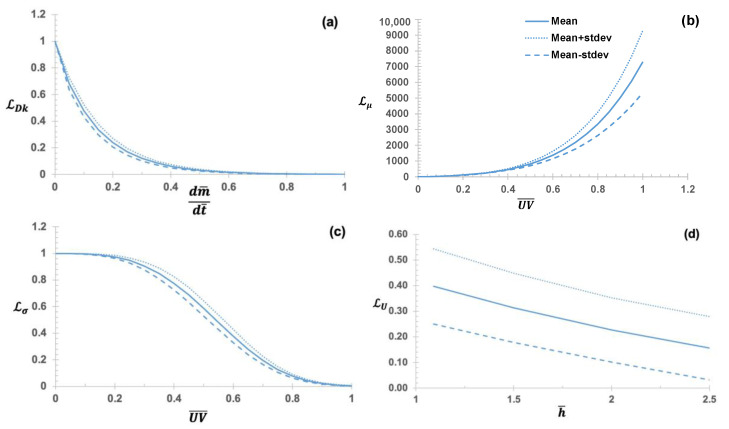
(**a**) Identified link function (LF) of diffusivity of air molecules into the PDMS nanopores, $L_{D_{K}}$($\frac{d\bar{m}}{d\bar{t}};\theta_{D_{K}}$) a function of the normalized air mass flux rate ($d\bar{m}/d\bar{t}$); (**b**) LF of the dynamic viscosity of NOA73 (pre-cured by UV), $L_{\mu }$($\bar{U_{v}};\theta_{\mu }$) a function of the normalized UV dose ($\bar{U_{v}}$); (**c**) LF of the surface tension of NOA73, pre-cured by U.UV $L_{\sigma \;}$($\bar{U_{v}};\theta_{\sigma }$); (**d**) LF of the potential (voltage) difference between the top surface of NOA73 and the lower surface of the PDMS template, $L_{U}$($\bar{h};\theta_{U}$) a function of the normalized height of the grating ($\bar{h}$). The dashed lines show a one-standard deviation range (i.e., ±σ) of the top 10 best-so-far rules.

**Figure 9 micromachines-14-01984-f009:**
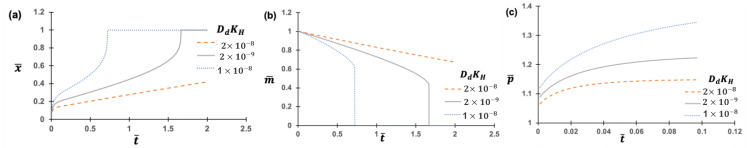
Parametric study of the impact of air diffusivity ${D_{d}K}_{H}$ [mol·s/kg]: (**a**) Normalized liquid height; (**b**) Normalized air mass; (**c**) Normalized air pressure. Other parameters are fixed, the dynamic viscosity μ=0.4 N·$s/m^{2}$, the surface tension $\sigma =0.04\;\frac{N}{m},and\;U_{0}=1\times 10^{-8}$.

**Figure 10 micromachines-14-01984-f010:**
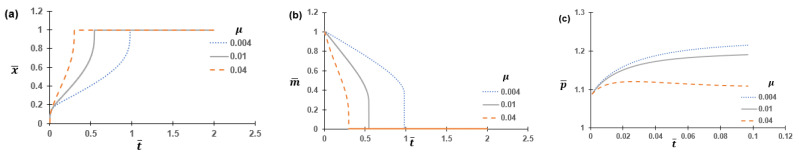
Parametric study of the impact of the dynamic viscosity μ [N·$s/m^{2}]$: (**a**) Normalized liquid height; (**b**) Normalized air mass; (**c**) Normalized air pressure. Other parameters are fixed, the diffusivity ${D_{d}K}_{H}=2\times 10^{-8}$ [mol·s/kg], surface tension $\sigma =0.04\;N/m,\;and\;U_{0}=1\times 10^{-8}$.

**Figure 11 micromachines-14-01984-f011:**
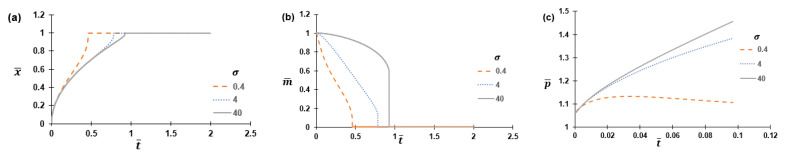
Parametric study of the impact of the surface tension σ [N/m]: (**a**) Normalized liquid height; (**b**) Normalized air mass; (**c**) Normalized air pressure. Other parameters are fixed, the diffusivity ${D_{d}K}_{H}=2\times 10^{-8}$ [mol·s/kg], dynamic viscosity $\mu =0.4\;N/m,\;and\;U_{0}=1\times 10^{-8}$.

**Figure 12 micromachines-14-01984-f012:**
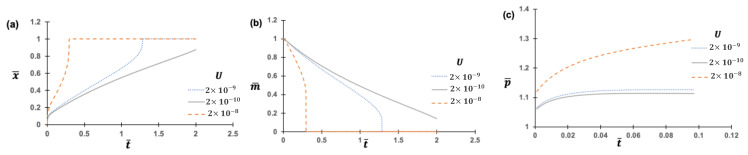
Parametric study of the impact of the initial potential (voltage) difference $U_{0}$: (**a**) Normalized liquid rise height; (**b**) Speed of normalized air mass reduction; (**c**) Normalized air pressure. Other parameters are fixed, the dynamic viscosity μ=0.4 N·$s/m^{2}$, the surface tension σ=0.04 N/m,and the diffusivity ${D_{d}K}_{H}=2\times 10^{-8}$ [mol·s/kg].

**Figure 13 micromachines-14-01984-f013:**
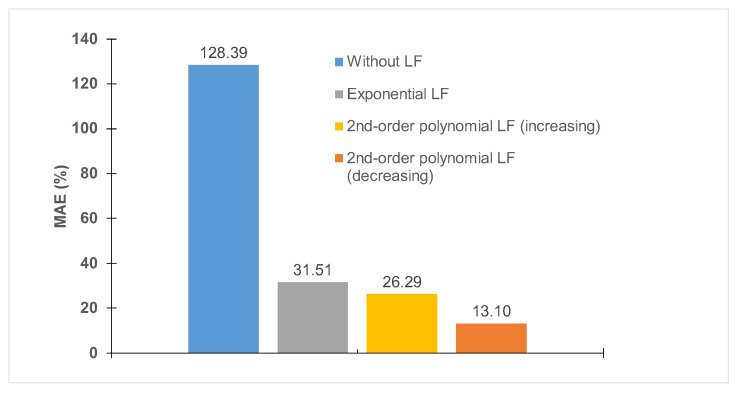
MAE of four cases of LF considered: No LF to account for voltage and dielectric constant, considering an exponential LF, and second-order polynomial LF with increasing and reducing trend.

**Figure 14 micromachines-14-01984-f014:**
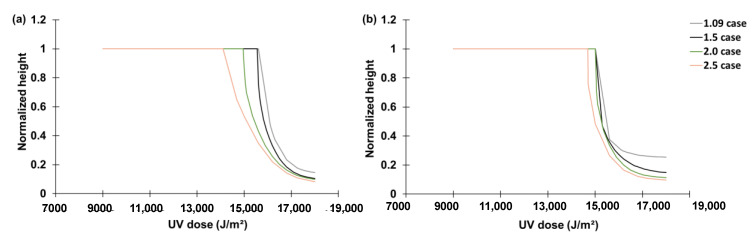
Comparison between (**a**) decreasing and (**b**) increasing trends of the electric potential link function $L_{U}$.

**Figure 15 micromachines-14-01984-f015:**
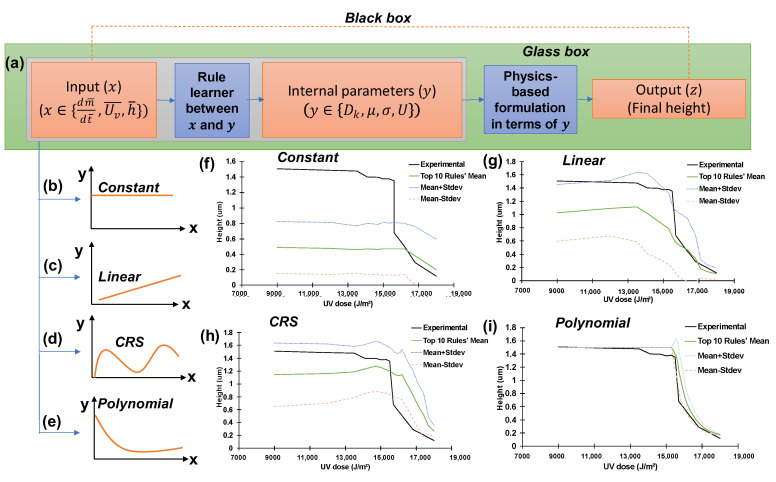
Illustration of the three global stages of the rule learning with diverse candidates for the hidden rules regarding important parameters ($y\in \{D_{k},\;\mu ,\;\sigma ,U\}$) in terms of descriptive variables ($x\in \{\frac{d\bar{m}}{d\bar{t}},\;\bar{U_{v}},\bar{h}\}$): (**a**) Overall learning framework (glass-box)—this step represents the entire process starting with input (experimental data and an assumed link function), moving to internal parameters, and ultimately resulting in an output (i.e., the final height attained by the liquid). The red dotted lines show a black-box approach that takes input (*x*) and produces output (z), without any understanding of underlying principles; (**b**) Constant model-based rule; (**c**) Linear model-based rule; (**d**) Complex nonlinear curve-based rule that can be learned by the cubic regression splines (CRS); (**e**) 2nd-order polynomial model-based rule; (**f**–**i**) Examples of the best-so-far predictions that use the constant model-based rules (**f**), linear model-based rules (**g**), CRS model-based rules (**h**) exhibiting poor performance, and the best-so-far prediction using 2nd-order polynomial model-based rules (**i**).

**Table 1 micromachines-14-01984-t001:** Summary of physical phenomena and the LF used to identify them.

Unknown Physics Term	Function Argument	Physical Meaning	Best Identified LF
$D_{d}K_{H}$	$\frac{d\bar{m}}{d\bar{t}}$	Diffusivity of trapped air into PDMS pores	2-parameter exponential function $(\exp(c_{1}x^{c_{2}}))\;$
μ	$\bar{U_{V}}\;$	Dynamic viscosity of the liquid	2-parameter exponential function $(\exp(c_{1}x^{c_{2}}))\;$
σ	$\bar{U_{V}}$	Surface tension of the liquid	2-parameter exponential function $(\exp(c_{1}x^{c_{2}}))\;$
U	$\bar{h}$	Potential (voltage) difference between the lower surface of PDMS and the upper surface of the liquid and dielectric constants of the materials	3-parameter 2nd-order polynomial function $(c_{1}x^{2}+c_{2}x+c_{3})\;$

## Data Availability

All experimental data are available upon request to the corresponding author.

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
