# Peer review of "Tackling Multi-Physics Nano-Scale Phenomena in Capillary Force Lithography with Small Data by Hybrid Intelligence"

_micromachines, 2023, doi:10.3390/mi14111984_

Round 1
Reviewer 1 Report
Comments and Suggestions for Authors
attached

Reviewer 2 Report
Comments and Suggestions for Authors
See the attached pdf file.

English is very good (albeit not free of minor mistakes and omissions). The authors should carefully re-read the manuscript to eliminate minor mistakes (e.g. um as notation for micrometer, inconsistencies in the use of articles, etc.); some incorrectly spelled words seem to point out that spellchecking was missing or incomplete (e.g. “understaning”, line 378). Such typos even appear in reference titles (e.g. [15] title has 2 typos, “Persuit” and “irredularity”).
Reviewer 3 Report
Comments and Suggestions for Authors
Jump in nano-grating is a physical phenomenon, which can be found in many places. This work develops an approach to understand this, which is interesting for many readers. I recommend this work to be considered into publication, although I am not familiar with the detailed content of this paper.
